# Alteration of Gut Immunity and Microbiome in Mixed Granulocytic Asthma

**DOI:** 10.3390/biomedicines10112946

**Published:** 2022-11-16

**Authors:** Bon-Hee Gu, Chae-Yun Rim, Sangjin Lee, Tae-Yong Kim, Sang-Seok Joo, Sang-Jin Lee, Han-Ki Park, Myunghoo Kim

**Affiliations:** 1Life and Industry Convergence Research Institute, Pusan National University, Miryang 50463, Republic of Korea; 2Department of Animal Science, College of Natural Resources & Life Science, Pusan National University, Miryang 50463, Republic of Korea; 3Department of Internal Medicine, School of Medicine, Kyungpook National University, Daegu 41944, Republic of Korea; 4Division of Allergy and Clinical Immunology, Department of Internal Medicine, School of Medicine, Kyungpook National University, Kyungpook National University Chilgok Hospital, Daegu 41404, Republic of Korea

**Keywords:** mixed granulocytic asthma, gut microbiota, intestinal immune cells, microbial metabolism

## Abstract

Growing evidence suggests that there is an essential link between the gut and lungs. Asthma is a common chronic inflammatory disease and is considered a heterogeneous disease. While it has been documented that eosinophilic asthma affects gut immunity and the microbiome, the effect of other types of asthma on the gut environment has not been examined. In this study, we utilized an OVA/poly I:C-induced mixed granulocytic asthma model and found increased Tregs without significant changes in other inflammatory cells in the colon. Interestingly, an altered gut microbiome has been observed in a mixed granulocytic asthma model. We observed an increase in the relative abundance of *the Faecalibaculum* genus and Erysipelotrichaceae family, with a concomitant decrease in the relative abundance of *the genera Candidatus arthromitus* and *Streptococcus*. The altered gut microbiome leads to changes in the abundance of genes associated with microbial metabolism, such as glycolysis. We found that mixed granulocytic asthma mainly affects the gut microbial composition and metabolism, which may have important implications in the severity and development of asthma and gut immune homeostasis. This suggests that altered gut microbial metabolism may be a potential therapeutic target for patients with mixed granulocytic asthma.

## 1. Introduction

Asthma is a common chronic inflammatory disease that affects more than 350 million people worldwide [1]. It is now considered a heterogeneous disease based on phenotypes describing clinical characteristics and endotypes describing pathophysiological mechanisms. Therefore, many studies have classified asthma phenotypes, and dividing type 2 asthma and non-type 2 asthma is the most common clinical classification method. Type 2 asthma was well known in the past for mechanisms related to allergic asthma, but recently, it has been known about the importance of non-allergic pathway such as epithelial alarmins and group 2 innate lymphoid cells (ILC2s). Non-type 2 asthma includes neutrophilic inflammation and paucigranulocytic inflammation, and various mechanisms, such as the inflammasome pathway and metabolic dysfunction, are being studied [2,3]. However, type 2 inflammation and non-type 2 inflammation influence each other; therefore, it is more common to be mixed. For example, eosinophil activation induces ETosis and the formation of Charcot-Leyden crystals, leading to inflammasome and neutrophilic inflammation [4]. In addition, various stimuli associated with airway inflammation, such as viruses and fatty acids, can stimulate both eosinophilic and neutrophilic inflammation [5,6]. Therefore, it is more common for mixed inflammation to be intertwined in real-world patients. In addition, patients with mixed granulocytic asthma show more severe and persistent inflammation-related symptoms and frequent exacerbations despite the use of corticosteroids rather than neutrophilic asthma and eosinophilic asthma [7,8].

Growing evidence suggests the importance of the gut environment, which consists of host immune cells and microbiota, for human health and diseases. Proper interactions between the microbiota and host immune cells in the gut are important for immune system development and immune homeostasis in the airway [9]. For example, short-chain fatty acids (SCFAs) produced by commensal microbes enhance bone marrow hematopoiesis, resulting in an increase in dendritic cells (DCs) with impaired ability to induce Th2 cell differentiation and macrophages in the lung. This immune regulation by gut microbial metabolites has significant implications for the development of lung inflammation, reducing susceptibility to allergic airway inflammation, and increasing antiviral immunity against influenza in the lung [10,11]. Mice with cigarette smoke exposure-induced lung inflammation showed elevated intestinal inflammation, which was mainly driven by the Th17 cell–neutrophil response [12]. However, critical questions remain as to how lung immunity alters gut immunity and vice versa.

Epidemiological studies have reported the presence of comorbidities between airway and intestinal inflammatory diseases. Considering that the lung and gut are both mucosal tissues and originate from identical embryonic organs, pulmonary and intestinal crosstalk needs to be elucidated in detail for more efficient treatment. The prevalence of inflammatory bowel disease (IBD) is significantly increased in patients with asthma, and the incidence of asthma is higher in IBD patients [13,14,15,16,17]. Therefore, researchers have attempted to understand the interaction between asthma and the gut environment in terms of immune cells and the microbiome to develop better therapeutics to limit the progression of diseases in both mucosal tissues. However, most previous studies have focused only on lung tissue and systemic immunity and not on the gut tissue environment in asthma. Moreover, most studies have been limited to allergic asthma, which is dominated by eosinophils. It is currently unknown how other types of asthma such as mixed granulocytic asthma affect the gut environment. Thus, in the current study, we aimed to determine whether mixed granulocytic asthma influences the gut microbiome and immune cell deposition. We found that OVA/poly I:C-induced mixed granulocytic asthma increased regulatory T cell deposition in the colon and altered the composition of the gut microbiome, which is related to host metabolism. These results may provide useful information for the development of effective therapeutic strategies in patients with mixed granulocytic asthma. 

## 2. Materials and Methods

### 2.1. Mice

Male C57BL/6 mice (6–7 weeks of age) were purchased from Central Lab Animal Inc. (Seoul, Republic of Korea), and all mice used in our experiments were bred at our facilities. All mice were bred in a specific pathogen-free (SPF) at the Pusan National University Laboratory Animal Resources Center. The SPF C57BL/6 mice used in this study were handled in the conventional room of Pusan National University. The mice were allowed access to food and water ad libitum. All animal experiments were performed according to the National Institutes of Health Guide for the Care and Use of Laboratory Animals and protocols approved by the Pusan National University-Institutional Animal Care and Use Committee (PNU-IACUC; approval no. PNU-2022-0172).

### 2.2. OVA/Poly I:C Induced Mixed Granulocytic Asthma Mouse Model

The mice were anesthetized with isoflurane and sensitized by intranasal inhalation (i.n.) of 50 μg OVA (Sigma-Aldrich, St. Louis, MO, USA) mixed with 10 μg poly I:C (Calbiochem, Darmstadt, Germany) on days 0, 1, 2, and 7. After 21 days, mice were re-challenged with OVA for four days (days 21, 23, 28, and 30) by intranasal inhalation (i.n.). The procedure was carried out by injecting 50 μg OVA in PBS into individual nasal passages. All the mice were sacrificed 24 h after the last OVA challenge (day 30). 

For analysis of lung inflammation, bronchoalveolar lavage (BAL) fluid was collected by instilling and withdrawing 0.8 mL of cold sterile PBS twice through the trachea. Total cell counts were obtained using a hemocytometer and stained with Diff-Quick (Sysmax, Kobe, Japan). Dissected lung tissue was used to prepare single-cell suspensions, and histopathological studies were performed using hematoxylin and eosin (H&E) staining. The single-cell suspensions were cultured overnight at 37 °C, and the supernatant was collected as lung homogenate for IL-4 (cat#431104, BioLegend, San Diego, CA, USA), IFN-γ (cat#430804, BioLegend, San Diego, CA, USA), and IL-17A (cat#432504, BioLegend, San Diego, CA, USA) ELISA according to the manufacturer’s instructions.

### 2.3. Isolation of Gut Lamina Propria Cells

Colonic lamina propria cells were isolated, as previously described [18]. Briefly, colon tissues were washed in PBS containing 1 mM DL-dithiothreitol (DTT; Sigma-Aldrich, Irvine, UK), 30 mM ethylenediaminetetraacetic acid (EDTA; Thermo Fisher Scientific/Ambion, Waltham, MA, USA), and 10 mM 4-[2-hydroxyethyl]-1-piperazineerhanesulfonic acid (HEPES; Thermo Fisher, Waltham, MA, USA) at 37 °C for 10 min. The tissues were washed again in PBS containing 30 mM EDTA and 10 mM HEPS at 37 °C for 10 min. After washing, the tissues were digested in RPMI 1640 containing 0.5 mg/mL collagenase VIII (Sigma-Aldrich, Chem, Fort Lauderdale, FL, USA) and 15 ug/mL DNase I (Sigma Aldrich; 90 mg/mL) at 37 °C for 1 h. The cell suspensions from the enzyme digestion were then applied to a Percoll (GE Healthcared/Amersham, Bucking-hampshire, UK) gradient (for lymphocytes: 40% Percoll on the top and 80% Percoll on the bottom) by centrifugation without breaking at room temperature.

### 2.4. Isolation of Gut mRNA and Quantitative PCR

Gut samples were treated with TRIzol™ Reagent (Thermo Fisher Scientific, Waltham, MA, USA) and mRNA was extracted with chloroform (Sigma Aldrich, Chem, USA), precipitated in isopropanol (Biosesang, Seoul, Republic of Korea), and washed in 75% alcohol (Bioseang, Seoul, Republic of Korea). cDNA was synthesized using AccuPower^®^ RT PreMix (Bioneer, Daejeon, Republic of Korea) in accordance with the manufacturer’s instructions. qRT-PCR was performed using a QuantStudio 1 Real-Time PCR system (Applied Biosystems, Waltham, CA, USA) with reaction conditions as follows; 50 °C for 2 min, 95 °C for 15 min, 95 °C for 20 s, 60 °C for 40 s, and 72 °C for 20 s (40 cycles), followed by melting curve analysis. The following primers for the analyzed genes were purchased from Applied Bionics (Seoul, Republic of Korea): *GAPDH* (forward primer 5′-ATC CTGCAC CAC CAA CTG CT-3′ and reverse primer 5′-GGG CCA TCC ACA GTC TTC TG-3′), *IL-10* (forward primer 5′-CAT CAT GTA TGC TTC TAT GCA G-3′ and reverse primer 5′-CCA GCT GGA CAA CAT ACT GCT-3′), *occludin* (forward primer 5′-CCT TCT GCT TCA TCG CTT CCT TA-3′ and reverse primer 5′-CGT CGG GTT CAC TCC CAT TAT-3′), *ZO-1* (forward primer 5′-GCC GCT AAG AGC ACA GCA A-3′ and reverse primer 5′-TCC CCA CTC TGA AAA TGA GGA-3′), *MUC2* (forward primer 5′-CCG ACT TCA ACC CAA GTG AT-3′ and reverse primer 5′-GAG CAA GGG ACT CTG GTC TG-3′), *LGR5* (forward primer 5′-ATT CGG TGC ATT TAG CTT GG-3′ and reverse primer 5′-CGA ACA CCT GCG TGA ATA TG-3′), *CHGA* (forward primer 5′-AAG GTG ATG AAG TGC GTC CT-3′ and reverse primer 5′-GGT GTC GCA GGA TAG AGA GG-3′), *Vil* (forward primer 5′-TCA AAG GCT CTC TCA ACA TCA C-3′ and reverse primer 5′-AGC AGT CAC CAT CGA AGA AGC-3′), and *Lyz1* (forward primer 5′-GTC ACA CTT CCT CGC TTT CC-3′ and reverse primer 5′-TGG CTT TGC TGA CTG ACA AG-3′).

### 2.5. Immunophenotyping Using Flow Cytometry

Flow cytometric analysis was performed on a Canto II flow cytometer (BD Biosciences) using FlowJo software v10.7.1 (Tree Star Inc., Ashland, OR, USA). Dead cells were excluded using a live/dead fixable dead cell stain (Thermo Fisher Scientific, Waltham, MA, USA). The following mouse-specific antibodies were used for staining: CD3 (145-2C11, BioLegend, San Diego, CA, USA), CD4 (RM4-5, BioLegend), Foxp3 (FJK-16s, Invitrogen), T-bet (4B10, Invitrogen, Eugene, OR, USA), RORγt (B2D, Invitrogen), GATA3 (16E10A23, BioLegend), Siglec F (S17007L, Biolegend), Ly6G (1A8, Biolegend), CD11b (M1/70, BioLegend), B220 (RA3-6B2, BioLegend), IgA (C10-3, Biolegend), and IgM (RMM-1, Biolegend).

### 2.6. 16s rRNA Gene Amplification and Sequencing

Fecal samples were kept frozen at −80 °C until further processing. DNA was extracted using the DNeasy PowerSoil Kit (Cat. No. 12855, Qiagen, Hilden, Germany) according to the manufacturer’s protocol. The extracted DNA was quantified using a Quant-IT PicoGreen (Invitrogen). Sequencing libraries were prepared according to Illumina 16S Metagenomic Sequencing Library protocols to amplify the V3 and V4 regions. The input gDNA 2ng was PCR amplified with 5× reaction buffer, 1 mM dNTP mix, 500 nM each of the general F/R PCR primers, and Herculase II fusion DNA polymerase (Agilent Technologies, Santa Clara, CA, USA). The general primer pair with Illumina adapter overhang sequences used for the first amplification was as follows: V3-F:5′-TCGTCGGCAGCGTCAGATGTGTATAAGAGACAGCCTACGGGNGGCWGCAG-3′, V4-R:5′-GT-CTCGTGGGCTCGGAGATGTGTATAAGAGACAGGACTACHVGGGTATCTAATCC-3′. The 1st PCR product was purified using AMPure beads (Agencourt Bioscience). Following purification, the 2 μL of 1st PCR product was PCR amplified for final library construction containing the index using NexteraXT Indexed Primer. The PCR products were purified using AMPure beads. The final purified product was then quantified using qPCR according to the qPCR Quantification Protocol Guide (KAPA Library Quantification kits for Illumina Sequencing platforms) and qualified using TapeStation D1000 ScreenTape (Agilent Technologies, Waldbronn, Germany). Paired-end (2 × 300 bp) sequencing was performed by Macrogen using the MiSeq platform (Illumina, San Diego, CA, USA). For microbial identification, FASTQ files for each sample were generated from the raw MiSeq sequence data. Sequences with a quality score over 25 were eliminated in paired-end reads in the FASTQ files, and F/R primer and chimera sequences were cut using DADA2 (v1.22.0). forward sequence 280 bp and reverse sequence 202 bp were removed, and the amplicon sequence variant (ASV) was generated. In addition, for the comparative analysis of microbial clusters, the QIIME2 (v2022.02) program was used to perform subsampling and normalization based on the number of reads of samples with a minimum number of reads among all samples. Sequences from each ASV were subjected to BLAST matching to the Reference DB (Silva 138) and assigned taxonomy information for the organism of the subject with the highest similarity.

To analyze the microbiome data, ASVs abundance and taxonomic information were utilized for the analysis of various microbial clusters using QIIME2. Alpha diversity was confirmed through the Shannon index and Chao1 index to confirm the species diversity and uniformity of the microbial community in the sample. Beta diversity between samples was confirmed through weighted UniFrac distance, and the relationship between samples was visualized using PCoA. The functional gene content of the microbes was classified based on the Greengenes (v.13.5) database and predicted using PICRUSt 2. Classified and predicted genes were normalized by 16S rDNA copy number, and metagenomic information was hierarchically clustered and classified based on the Kyoto Gene and Genome Encyclopedia (KEGG) database. The LEfSe method was used to identify the difference between normal and NID pathways. The threshold of the Kruskal–Wallis test was 0.05, and the algebraic linear discriminant analysis score (LDA) was ≥2.

## 3. Results

### 3.1. Establishment of the OVA/Poly I:C Model of Mixed Granulocytic Asthma

To investigate the influence of mixed granulocytic asthma on the gut, we established an OVA/poly I:C mouse model by sensitization with the widely used allergen OVA for eosinophilic asthma and the viral mimetic poly I:C to activate neutrophils, followed by OVA challenge (Figure 1A). Histologically, the lungs of the OVA/poly I:C model mice showed inflammation with infiltration of inflammatory cells (Figure 1B). Bronchoscopy and bronchoalveolar lavage (BAL) of the OVA/poly I:C model showed that both eosinophils and neutrophils were significantly increased in addition to the total cell numbers (Figure 1C). Considering that the OVA-induced eosinophilic asthma model showed an increase in eosinophils without neutrophil accumulation, these results suggest that the OVA/poly I:C model elicited both eosinophil- and neutrophil-mediated airway inflammation, corresponding to the cellular characteristics detected in the mixed granulocytic asthma phenotype. Th2 cytokines (IL-4, IL-5, and IL-13) are well-known inflammatory mediators in eosinophilic asthma, and Th1 and Th17 related cytokines are implicated in the pathobiology of non-eosinophilic asthma, with increased neutrophils, neutrophilic asthma, and mixed granulocytic asthma [19]. Thus, we assessed cytokine levels in BALF and lung homogenates to identify inflammatory mediators in OVA/poly I:C-induced mixed granulocytic asthmatic mice. Mice in the OVA/poly I:C model released highly elevated levels of IFN-γ and slightly higher levels of IL-4 (Figure 1D). However, IL-17A levels did not differ between control and asthmatic animals. These results suggest that the OVA/poly I:C model is a mixed granulocytic asthma model related with immune response of type I immunity. 

### 3.2. Increased Regulatory T Cells in the Gut with Mixed Granulocytic Asthma 

We performed immunophenotyping of the gut by focusing on granulocytes, T cells, and B cells to determine the effects of lung inflammation on gut immune homeostasis in an OVA/poly I:C-induced mixed granulocytic asthma model. In contrast to the lungs, the OVA/poly I:C model did not influence the deposition of eosinophils and neutrophils in the colon (Figure 2A,B). However, we found that Foxp3 expression in CD4^+^ T cells increased in the colonic lamina propria of the OVA/poly I:C model, mainly in RORγt^−^Foxp3^+^CD4^+^ T cells (Figure 2C and Appendix A). However, IL-10 expression in the colon did not differ between the groups (Figure 2D). Other types of T cells, such as RORγt−, T-bet, or GATA3 expressing CD4^+^ T cells, did not change (Figure 2C). In addition, no difference was observed in B cells and expression of genes for tight junction and intestinal epithelial cell markers regarding tissue damage or inflammation in the colon (Figure 2E and Appendix A).

### 3.3. Altered Gut Microbiome in Mixed Granulocytic Asthma 

To examine whether mixed granulocytic asthma in the airways influences changes in gut microbial populations, we compared the gut microbial diversity and composition in fecal samples obtained from three independent experiments. There were no significant differences in alpha and beta diversities among the groups (Figure 3A,B). In the bacterial taxa analysis, we did not observe significant changes in bacterial composition at the phylum level. However, significant changes were detected at both the genus and family levels. Interestingly, OVA/poly I:C-induced mixed granulocytic asthma mice showed changes in the phylum Firmicutes only. Clostridiaceae and Streptococcaceae families of the Firmicutes phylum were significantly reduced, and the Erypsipelotrichaceae family was significantly increased in the OVA/poly I:C model. Similarly to the family level, OVA/poly I:C mice displayed a significant decrease in *Candidatus arthromitus* and *Streptococcus* at the genus level (Figure 3C,D). The relative abundance of the genus *Faecalibaculum*, belonging to Erypsiplelotrichaceae, was increased in OVA/poly I:C-induced mixed granulocytic asthma. In addition, *the NK4A214 group* at the genus level, belonging to the Oscillospiraceae family of the Firmicutes phylum, was increased in the model (Figure 3C,D). 

### 3.4. Changes of Functional Gene Abundance in Gut Microbiome of Mixed Granulocytic Asthma 

Next, we investigated changes in functional gene abundance within the gut microbiome using PICRUst 2 analysis. There were 25 types of functions that differed significantly between the control and OVA/poly I:C-induced mixed granulocytic asthma mice, of which 15 functional levels were reduced and 10 were increased. The genes associated with carbohydrate metabolism pathways, namely glycolysis, the pentose phosphate pathway, and the tricarboxylic acid (TCA) cycle, were increased in the fecal microbiome of the OVA/poly I:C model. In contrast, genes associated with the purine and fatty acid biosynthesis pathways were decreased in the fecal microbiome of the OVA/poly I:C model (Figure 4). These results implied that microbial metabolism is influenced by the development of mixed granulocytic asthma. 

## 4. Discussion

Although growing evidence shows an interaction between the gut environment and the pathophysiology of asthma, most previous studies have focused on eosinophilic asthma, and there is no study in terms of mixed granulocytic asthma. This is the first report to show the effect of mixed granulocytic asthma on the gut environment, including gut immunity and microbiome, using immunophenotyping of immune cells in colon tissue and metagenomic analysis of fecal samples. We established an OVA/poly I:C mouse model in which both neutrophils and eosinophils were increased in the lungs. Moreover, the model showed increased immune response of type 1 immunity, with increased levels of IFN-γ in the lungs. Clinical studies have reported an association between IFN-γ and asthma severity [20,21,22,23]. Therefore, although further human clinical studies are needed to verify the results shown here, our mouse model in the present study may imply the influence of severe asthma accompanied by type 1 immunity involved mixed granulocytic phenotype on the gut environment.

Human and murine studies have reported an association between gut microbiota and asthma, mainly with eosinophilic asthma phenotypes. The decrease in short-chain fatty acid (SCFAs)-producing bacteria, including *Lactobacillus* and *Bifidobacterium,* in the gut has received much attention in eosinophilic asthma. In a human birth cohort study of fecal microbiology, children with a decrease in the relative abundance of *Bifidobacterium* had a higher risk of developing asthma [24,25]. Oral supplementation with *Lactobacillus* and *Bifidobacterium,* metabolite SCFAs, and soluble fiber fermented to SCFAs alleviated eosinophilic asthma [26]. We also found that lung inflammation, defined as mixed granulocytes, changed the gut microbiome with substantial differences in microbial composition compared with control animals. Despite the increased frequency of regulatory T cells in the colons of mice with mixed granulocytic asthma, there was no difference in the relative abundance of SCFA-producing bacteria in our study. The major end product of microbial fermentation, SCFAs, are known to induce regulatory T cell (Treg cell) differentiation via increased histone acetylation or activation of the SCFA receptor GPR41 or GPR43. In line with the lack of changes in SCFA-producing bacteria, mixed granulocytic asthma did not show a decreased Treg population or IL-10 expression in the colon. In fact, we observed a slight increase in the proportion of Treg cells (CD4^+^Foxp3^+^ T cells) in the colon following the development of mixed granulocytic asthma without tissue damage or inflammation. A major increase was observed in RORγt^−^Foxp3^+^CD4^+^ T cells. Considering that the most of IL-10 producing Treg cells are RORγt^+^Foxp3^+^CD4^+^ T cells maintained and induced by microbial antigens [27,28,29], these results suggest that changes in colonic Tregs are induced by host immune factors rather than by gut microbial factors. Although the reason for the increased Treg count in the colon of mice with mixed granulocytic asthma should be identified in detail in the future, systemic environmental changes by lung inflammation such as cytokine milieu or function of circulating immune cells may be associated with the changed colonic immune environment. Additionally, intestinal Foxp3^+^ Treg cells constitute a group of distinct subsets, which differ developmental origins (thymus-derived Treg or peripherally derived Treg) and have functional heterogeneity and plasticity depending on intestinal tissue microenvironment. Thus, we need to figure out what kinds of Treg cell were increased in the gut of mice with mixed granulocytic asthma to understand the biological relevance. Although we did not observe changes in gut pro-inflammatory immune cells such as Th1, Th17, and neutrophils in the mixed granulocytic asthma model, we did not determine whether gut inflammation and immune responses were altered in the presence of mixed granulocytic asthma in the context of gut inflammation. Because these data imply that mixed granulocytic asthma change the systemic immune system leading to intestinal immune microenvironmental change without direct tissue damage or inflammation, gut inflammation models, such as DSS-induced colitis models, can be applied in future studies. 

Our data showed that OVA/poly I:C-induced mixed granulocytic asthma in mice increased the relative abundance of *the Faecalibaculum* genus and Erysipelotrichaceae family, with a concomitant decrease in the relative abundance of *the genera Candidatus Arthromitus* and *Streptococcus*. Erysipelotrichaceae is known to be a highly immunogenic bacterium that is enriched in antibiotic-induced gut dysbiosis and positively correlates with the level of tumor necrosis factor alpha (TNF-α) [30,31,32]. The taxon in Erysipelotrichaceae was discovered as a highly IgA-coated colitogenic bacterium [31]. Notably, a recent study showed that *Faecalibaculum Rodentium,* the only species in *the genus Fecalibaculum*, alters intestinal epithelial homeostasis by suppressing the production of retinoic acid, which supports eosinophil survival and consequently inhibits epithelial turnover by IFN-γ [33]. Furthermore, Erysipelotrichaceae is highly associated with host lipid metabolism. Erysipelotrichaceae was highly enriched in obese individuals and high-fat or Western diet-induced obese mice [34,35,36]. The correlation between host cholesterol metabolism and the abundance of Erysipelotrichaceae has been observed in human and animal models of hypercholesterolemia, which show a decrease in abundance due to improvements in cholesterol metabolism [37,38]. *Candidatus arthromitus,* another altered microbial bacterium in mixed granulocytic asthma, is also involved in the modulation of host metabolism. The reduction of *Candidatus arthromitus* has been detected in high-fat diet-induced obesity [37,39]. Genetically manipulated metabolically resistant mice, even in high-fat diet feeding, showed a higher abundance of *Candidatus arthromitus* compared to normal mice with high-fat diet-induced metabolic diseases [40]. Therefore, mixed granulocytic asthma might influence host cholesterol metabolism, and further investigation is needed to verify these mechanisms in detail. 

Altered gut microbiomes usually lead to changes in the gut microbial fermentation. In the present study, several microbial pathways were found to be altered by mixed granulocyte asthma. Interestingly, KEGG pathway analysis revealed that most of the changes in microbial pathways were metabolic pathways. For example, a mixed granulocyte asthma model showed a decreased abundance of genes for 5-Aminoimidazole ribonucleotide biosynthesis and L-lysine biosynthesis, but an increased abundance of genes associated with the pentose phosphate pathway and glycolysis I/II. Therefore, mixed granulocytic asthma may influence intestinal glucose metabolism, and further investigation to understand the functional role of this metabolic change is needed to verify the mechanisms in detail. Understanding this information may help develop new strategies to improve the efficacy of medication for patients with mixed granulocyte asthma. 

Although this study sheds light on the effect of mixed granulocyte asthma development on gut immunity and the microbiome, it has some limitations. Each set of mice experiments showed a differential baseline in microbiota composition; thus, we combined three different sets of mice experiments to distinguish the changed microbial taxa between groups. In addition, the relative abundance (%) of the significantly altered microbial taxa was less than 3%. Therefore, we must carefully consider the meaning of minor compositional alterations. Other limitations to interpret the influence of lung inflammation on the gut microbiota and immunity shown here will be health condition of mice and challenge route. General health condition of mice in disease model can affect feeding behavior leading to weight loss or food intake reduction that can affect gut immune environment, particularly gut microbiome. In general, it is normal phenomenon in the most of disease development, which is hard to exclude in animal study. Thus, even if the changed health condition is natural, we always need to be careful to interpret results of gut environmental change in various disease model. Additionally, reagents administrated via intranasal challenge may enter esophagus and affect the gut environment directly, thus intratracheal challenge will be better than intranasal challenge to exclude possibility of oral ingestion for lung-gut axis study. 

In summary, the mixed granulocyte asthma model showed distinct gut microbiome changes, which were characterized by an increased relative abundance of *the Faecalibaculum* genus and Erysipelotrichaceae family with a concomitant decrease in the relative abundance of *the genera Candidatus Arthromitus* and *Streptococcus*. Gut microbial metabolism, including glycolysis, is also altered by the development of mixed granulocyte asthma. These results suggest that altered gut microbial metabolism is a potential therapeutic target for patients with mixed granulocyte asthma. 

## Figures and Tables

**Figure 1 biomedicines-10-02946-f001:**
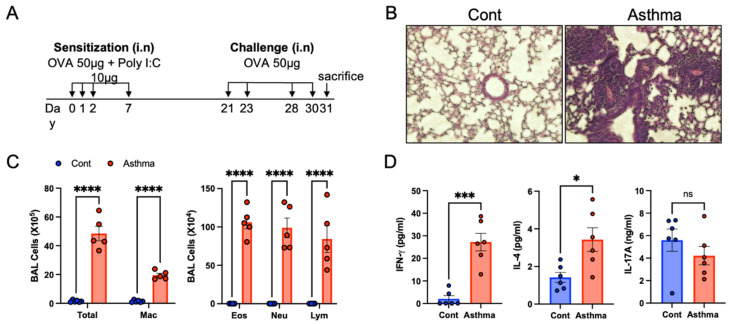
Establishment of the OVA/poly I:C model of mixed granulocytic asthma. (**A**) Schematics diagram of the OVA/poly I:C model. Mice were sensitized with OVA and poly I:C and then challenged with OVA as shown. (**B**) Representative images of hematoxylin and eosin stained lung section (original magnification: X200). (**C**) Differential cell counts in BAL fluid showing total cells, macrophages (Mac), eosinophils (Eos), neutrophils (Neu), and lymphocytes (Lym). (**D**) Levels of cytokine in whole lung homogenates of mice. Data shown are represented as mean ± SEM and representative of 3 independent experiments with 5 to 7 mice in each group. * *p* < 0.05, *** *p* < 0.001, and **** *p* < 0.0001, as determined by Student’s *t* test.

**Figure 2 biomedicines-10-02946-f002:**
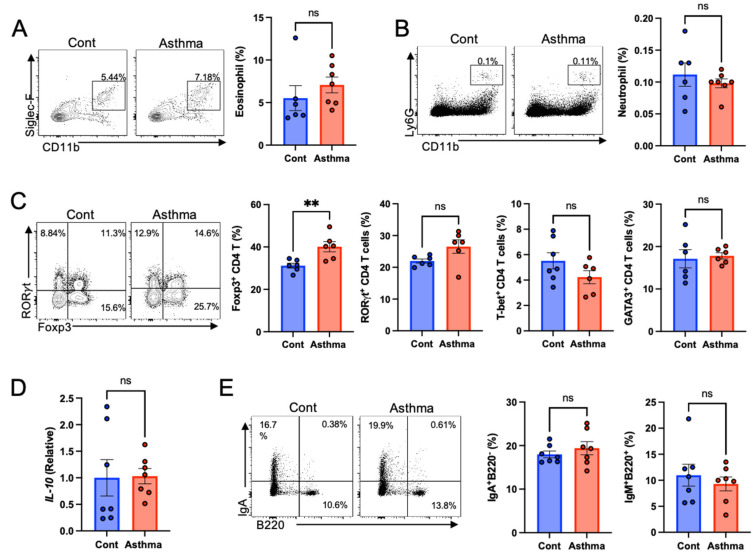
Increased regulatory T cells in the gut of mixed granulocytic asthma. Representative and cumulative flow cytometric analyses of (**A**) CD11b^+^ SiglecF^+^ eosinophils after neutrophil exclusion and (**B**) CD11b^+^ Ly6G^+^ neutrophils in the colon of the indicated group of mice. (**C**) Representative and cumulative flow cytometry analyses of Foxp3, RORγt, T-bet, or GATA3 expressing CD4^+^ T cell subsets gated on CD3 in the colon of the indicated group of mice. (**D**) Relative IL-10 expression in the colon. (**E**) Representative and cumulative flow cytometry analyses of IgA^+^B220^−^ and IgM^+^B220^+^ B cells in the colon of mice in the indicated groups. Data are represented as mean ± SEM and are representative of three independent experiments with 5 to 7 mice in each group. ** *p* < 0.01 as determined by Student’s *t* test.

**Figure 3 biomedicines-10-02946-f003:**
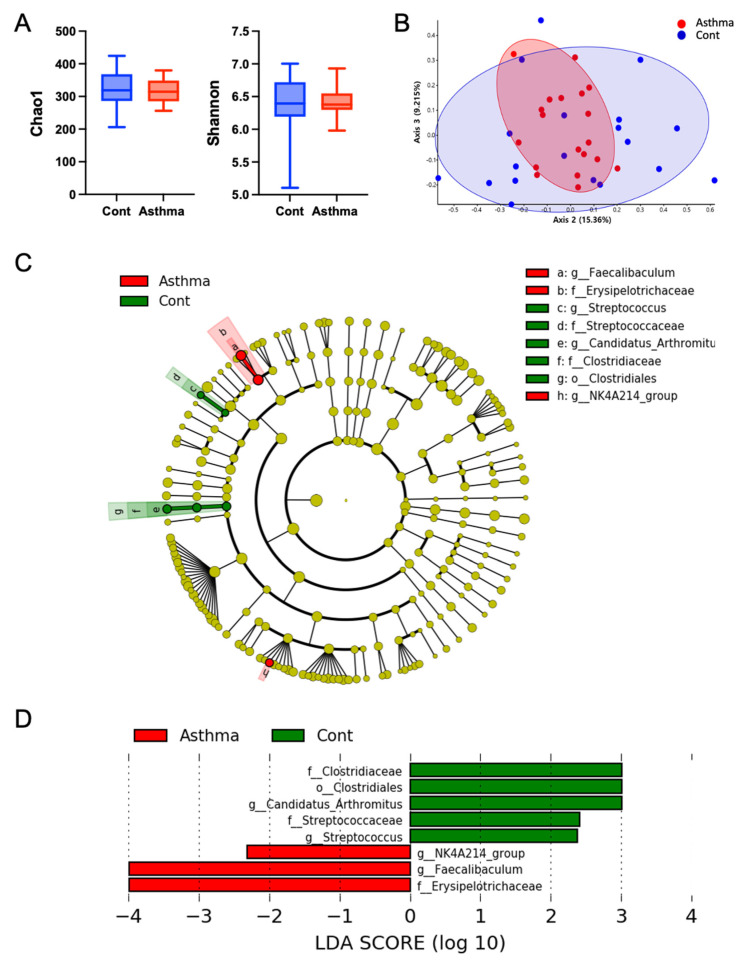
Altered gut microbiome in mixed granulocytic asthma. (**A**) Chao1 and Shanon index for alpha diversity. (**B**) PCoA plot by Brady-Cutis distance. (**C**) Taxanomic cladogram generated by LEfSe showing control enriched taxa (green) and mixed granulocytic asthma enriched taxa (red). (**D**).

**Figure 4 biomedicines-10-02946-f004:**
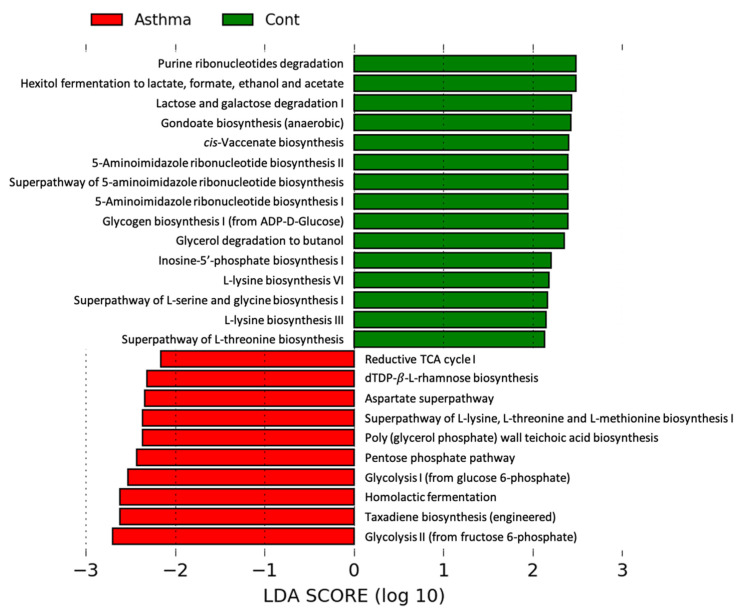
Functional changes within the gut microbiome in mixed granulocytic asthma. Functional pathways enriched in each groups based on PICRUSt prediction. LEfSe analysis (LDA > 2, *p* < 0.05).

## Data Availability

Not applicable.

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
