# Peer review of "Alteration of Gut Immunity and Microbiome in Mixed Granulocytic Asthma"

_biomedicines, 2022, doi:10.3390/biomedicines10112946_

Round 1

Reviewer 1 Report

In this manuscript, Gu and Rim et al. use a OVA/Poly I:C induced model of experimental mixed granulocytic asthma to determine if this airway disease may induce changes in gut microbiome and immune landscape. The authors show minor alterations in both these compartments. Overall the manuscript has a clear rationale, is concise and well written, with very detailed methods content that was easy to follow. The presentation of data was logical and hence a pleasure to read. However, there are a few concerns that need to be addressed here for the authors to support their conclusions.

1. The OVA/Poly I:C induced model of experimental mixed granulocytic asthma shows a strong increase in eosinophils and neutrophils but no data supporting the development of airway hyperreactivity has been shared. Given that bronchoconstriction is a major complication of asthma, this data is crucial to call this a model of asthma.

2. In Fig 1 and based on higher numbers of TH1 but not TH17 cells, the authors conclude that "the OVA/Poly I:C model of mixed granulocytic asthma is mediated by TH1 lymphocytes". The mere presence of a cell type should not be construed as a causative relationship. The authors may need to tone down this conclusion or provide direct evidence to support the causal relationship in this experimental model of asthma.

3. The authors identify minor differences in numbers of Tregs while none of the other cell types in the gut of asthmatic mice were affected. Hence, the biological relevance of this minor Treg perturbation is unclear. Indeed, IL10 levels was unaffected and none of the myeloid cells were affected. Did the authors see any differences in histopathology of the gut? If everything looked fine, can the authors discuss what the data might mean?

4. A major confounding factor in these studies is that the authors administered mice the sensitization and allergen challenge via intranasal route which does have a tendency to be swallowed and licked by the mice. Thus, eventually affecting the gut immune system and the microbiome. Whether the changes in gut microbiome and immunity are an effect of asthma or direct oral ingestion needs to be addressed and better controlled.

5. Induction of asthma make the mouse sick and the sickness can affect mouse feeding behavior which manifests itself as reduced food intake and weight loss. It is unclear if the subtle changes in the Tregs and microbiome composition may be due to less feeding and hence starvation induced perturbations (and may have anything to do with the asthma). This can be a major issue (that may have direct human relevance) but still needs to be controlled for, or at least directly addressed in the discussion as a major limitation of the model.  

6. No description for Fig 3B is given in the result section.

Reviewer 2 Report

The mixed granulocyte asthma model showed distinct gut microbiome changes, which were characterized by an increased relative abundance of the Faecalibaculum genus and Erysipelotrichaceae family with a concomitant decrease in the relative abundance of the genera Candidatus Arthromitus and Streptococcus.

Gut microbial metabolism, including glycolysis, is also altered by the development of mixed granulocyte asthma.

These results suggest that altered gut microbial metabolism is a potential therapeutic target for patients with mixed granulocyte asthma.

The relative abundance (%) of the significantly altered microbial taxa  was less than 3%. Therefore, we must carefully consider the meaning of minor compositional alterations.
